# Picrasidine J, a Dimeric β-Carboline-Type Alkaloid from *Picrasma quassioides*, Inhibits Metastasis of Head and Neck Squamous Cell Carcinoma

**DOI:** 10.3390/ijms241713230

**Published:** 2023-08-25

**Authors:** Hsin-Yu Ho, Chia-Chieh Lin, Yu-Sheng Lo, Yi-Ching Chuang, Mosleh Mohammad Abomughaid, Ming-Ju Hsieh

**Affiliations:** 1Oral Cancer Research Center, Changhua Christian Hospital, Changhua 500, Taiwan; 183581@cch.org.tw (H.-Y.H.); 181327@cch.org.tw (C.-C.L.); 165304@cch.org.tw (Y.-S.L.); 177267@cch.org.tw (Y.-C.C.); 2Department of Medical Laboratory Sciences, College of Applied Medical Sciences, University of Bisha, Bisha 61922, Saudi Arabia; moslehali@ub.edu.sa; 3Doctoral Program in Tissue Engineering and Regenerative Medicine, College of Medicine, National Chung Hsing University, Taichung 402, Taiwan; 4Graduate Institute of Biomedical Sciences, China Medical University, Taichung 404, Taiwan

**Keywords:** picrasidine J, head and neck carcinoma, metastasis, EMT, KLK-10, MAPK signaling

## Abstract

Head and neck squamous cell carcinoma (HNSCC) are associated with recurrence, distant metastasis, and poor overall survival. This highlights the need for identifying potential therapeutics with minimal side-effects. The present study was designed to investigate the anticancer effects of picrasidine J, a dimeric β-carboline-type alkaloid isolated from the southern Asian plant *Picrasma quassioides.* The results showed that picrasidine J significantly inhibits HNSCC cell motility, migration, and invasion. Specifically, picrasidine J inhibited the EMT process by upregulating E-cadherin and ZO-1 and downregulating beta-catenin and Snail. Moreover, picrasidine J reduced the expression of the serine protease KLK-10. At the signaling level, the compound reduced the phosphorylation of ERK. All these factors collectively facilitated the inhibition of HNSCC metastasis with picrasidine J. Taken together, the study identifies picrasidine J as a potential anticancer compound of plant origin that might be used clinically to prevent the distant metastasis and progression of HNSCC.

## 1. Introduction

Head and neck squamous cell carcinoma (HNSCC) is the sixth most commonly detected malignancy worldwide. It originates from the mucosal epithelium in the oral cavity, pharynx, and larynx mostly due to excessive tobacco use and/or alcohol consumption [1]. Previous infection with human papillomavirus (HPV) is another potential cause of HNSCC, especially oropharyngeal carcinoma [2]. Clinically approved vaccines targeting most of the oncogenic viral strains are considered promising in preventing the development of HPV-positive HNSCCs [3]. Physical examinations to identify pre-malignant lesions in the oral cavity is the primary approach for the early detection of HNSCCs. However, in many cases, advanced-stage HNSCC develops without a clinical history of pre-malignant lesions. Because of this reason, most of the HNSCCs present with lymph node metastasis at the time of detection [4].

Surgery, radiotherapy, and chemotherapy, either separately or in combination, are considered gold standard therapeutic interventions for HNSCCs. However, about 40–60% of patients undergoing these treatments experience recurrence and develop resistance to subsequent treatments [5]. Regarding cancer prognosis, HPV-negative HNSCCs are associated with a higher metastatic rate and lower 5-year survival rate than HPV-positive HNSCCs [6]. The application of tri-modality therapy including surgery, high-dose cisplatin, and radiation has been found to improve disease-free survival even in highest-risk individuals. However, the therapy is known to have serious side-effects and increase the risk of non-cancer related mortality in patients [7,8]. Considering the molecular landscape and immune profiles of HNSCCs, cetuximab—a monoclonal antibody targeting epidermal growth factor receptor—and immune checkpoint inhibitors pembrolizumab and nivolumab have received clinical approval for the treatment of cisplatin-refractory recurrent or metastatic HNSCCs [9]. Immune checkpoint inhibitors exhibit higher treatment efficacy when used in combination with chemotherapy. However, in some patients, these inhibitors may increase the progression of cancer or induce immune-related complications [10,11].

Phytochemicals are naturally occurring plant products with many health benefits, including antioxidant, anticancer, and anti-inflammatory properties [12,13]. Phytochemicals exert anticancer effects through various mechanisms, including inactivation of carcinogens; suppression of oxidative stress; inhibition of cancer cell growth, proliferation, and metastasis; and modulation of immune and inflammatory responses [14,15]. One classical example of a natural plant product with anticancer properties is picrasidine, which is a dimeric β-carboline-type alkaloid isolated from the southern Asian plant *Picrasma quassioides* [16]. Some recent studies have shown that the compound (such as picrasidine C, G, or N) acts as an agonist of peroxisome proliferator-activated receptor (PPAR) or exerts anticancer effects [17,18,19]. Picrasidine I and J were first discovered by the team of Ohmoto, and they exhibit a similar structure [20]. Our previous studies have found that picrasidine I induces cell apoptosis in oral cancer cells and nasopharyngeal carcinoma cells [16,21]. However, the anticancer activity and the mechanism behind picrasidine J is not well understood. Considering above-mentioned anticancer properties, the present study was designed to assess the anticancer effects and mode of action of picrasidine J in HNSCC cell lines.

## 2. Results

### 2.1. Cytotoxic Effect of Picrasidine J in HNSCC Cells

To investigate whether picrasidine J affects the viability of HNSCC cells, two different cell lines were treated with different concentrations (25, 50, and 100 μM) of the compound for 24 h. The cells treated with DMSO were only used as controls. As observed in Figure 1B,C, the treatment of picrasidine J did not reduce the viability of HNSCC cells. Only one cell line showed a reduced viability after the treatment with highest concentration of picrasidine J (100 μM) (Figure 1B). Overall, these observations indicate that picrasidine J does not have any cytotoxic effect in HNSCC cells.

### 2.2. Effect of Picrasidine J on Migration and Invasion of HNSCC Cells

Since picrasidine J did not exhibit any cytotoxic effects, we next investigated it on HNSCC cell migration and invasion. Firstly, the cells were treated with different concentrations of picrasidine J for 24 h and subjected to a wound healing assay. The distance migrated by the cells was estimated at 3, 6, and 24 h. As observed in Figure 2A–D, all tested concentrations of picrasidine J caused a significant reduction in HNSCC cell motility at all timepoints. Given these observations, we next performed migration and invasion assays, as described in details in the Method and Material section. As observed in Figure 2E–H, the treatment with picrasidine J significantly reduced the number of both migrated and invaded HNSCC cells in a dose-dependent manner. Overall, these observations indicate that picrasidine J is capable of inhibiting the migration and invasion of HNSCC cells.

### 2.3. Effect of Picrasidine J on Epithelial–Mesenchymal Transition of HNSCC Cells

Epithelial–mesenchymal transition (EMT) is considered to be a hallmark process responsible for cancer metastasis. The process is characterized by the downregulation of epithelial markers (E-cadherin and cytokeratin) and upregulation of mesenchymal markers (N-cadherin, vimentin, fibronectin, and Snail) [22]. Given the significant involvement of EMT on cancer metastasis, we next investigated the effect of picrasidine J treatment on the expression of selective EMT markers. We assessed the expression of epithelial-specific markers, E-cadherin and ZO-1, and EMT activators, beta-catenin and Snail, in picrasidine J-treated HNSCC cells. As observed in Figure 3A–D, the picrasidine J treatment significantly increased the expression of E-cadherin and ZO-1 and reduced the expression of beta-catenin and Snail in both cell lines. This finding clearly indicates an induction of epithelial characteristics and inhibition of EMT in picrasidine J-treated cancer cells. Thus, picrasidine J is capable of inhibiting the EMT process in HNSCC cells, which might be a possible anti-metastatic mode of action of picrasidine J.

### 2.4. Effect of Picrasidine J on Kallikrein-10 Expression in HNSCC Cells

Kallikrein-10 (KLK-10) is a serine protease that is aberrantly expressed in HNSCC and plays a vital role in the cancer onset and progression [23,24,25,26]. In this study, we investigated the effects of picrasidine J treatment on the expression of KLK-10 in HNSCC cells. We also determined the involvement of this protease in picrasidine J-mediated inhibition of HNSCC metastasis. As observed in Figure 4A–C, the picrasidine J treatment significantly reduced the expression of KLK-10 in both cell lines in a dose-dependent manner. Given this observation, we performed a siRNA-mediated downregulation of KLK-10 expression in cells (Figure 4D,E) and executed a wound healing assay in the presence or absence of picrasidine J. As observed in Figure 4F–I, the treatment with both picrasidine J and siRNA caused a further reduction in cancer cell motility compared to that achieved by the picrasidine J treatment alone. Next, we performed migration and invasion assays in KLK-10-silenced HNSCC cells with or without picrasidine J treatment. As expected, the treatment with both picrasidine J and siRNA caused a further reduction in cancer cell metastasis and invasion compared to that achieved by the picrasidine J treatment alone (Figure 5A–D). Collectively, these observations indicate that picrasidine J inhibits HNSCC cell metastasis and invasion by downregulating the expression of serine protease KLK-10.

### 2.5. Picrasidine J Inhibits Cell Metastasis via ERK Signaling Pathway in HNSCC Cells

The MAPK signaling pathway plays a crucial role in mediating cancer metastasis [27]. To investigate the impact of picrasidine J on cellular signaling pathways, we determined the expression of MAPK pathway components (ERK, P38, and JNK) and AKT in HNSCC cells treated with picrasidine J. As observed in Figure 6A–C, the picrasidine J treatment significantly reduced the phosphorylation of ERK in both cell lines. However, no significant effect of picrasidine J was observed on other tested signaling components. Given the significant involvement of the ERK signaling pathway, we treated the cells with a well-established ERK inhibitor (U0126) and conducted wound healing, migration, and invasion assays in the presence and absence of picrasidine J. As observed in Figure 7A–D, the treatment with both picrasidine J and ERK inhibitor caused a further reduction in cancer cell motility compared to that caused by picrasidine J treatment alone. Similarly, a significant reduction in HNSCC migration and invasion was observed after picrasidine J and U0126 co-treatment compared to that after picrasidine J treatment alone (Figure 7E–H). Overall, these observations indicate that picrasidine J reduces HNSCC cell metastasis by downregulating the ERK signaling pathway.

## 3. Discussion

The present study describes the effect and mode of action of the plant-based compound picrasidine J on the invasion and metastasis of HNSCC. Picrasidine J is a dimeric β-carboline-type alkaloid isolated from the southern Asian plant *Picrasma quassioides*. Previous studies on other picrasidine derivatives have highlighted their potential anticancer activities [28]. In triple-negative breast cancer, picrasidine G has been found to reduce cancer cell proliferation by inhibiting the EGFR/STAT3 signaling pathway [19]. In oral squamous cell carcinoma, picrasidine I was found to exert cytotoxic effects by inducing cell cycle arrest and apoptosis [16]. Picrasidine I was also found to induce apoptosis in nasopharyngeal carcinoma by modulating heme oxygenase 1 via AKT and ERK signaling pathways [21]. In the present study, we identified anti-metastatic effects of picrasidine J in HNSCC (Figure 2). The compound has been found to prevent HNSCC cell migration and invasion by inhibiting EMT (Figure 3), downregulating KLK-10 (Figure 4 and Figure 5), and suppressing ERK phosphorylation (Figure 6 and Figure 7). However, we could not observe any cytotoxic effect of picrasidine J in HNSCC cells (Figure 1).

One recent study investigating the anticancer effects of phytochemicals has demonstrated that dehydrocrenatidine, another β-carboline-type alkaloid isolated from *Picrasma quassioides*, significantly reduces the migration and invasion of head and neck cancer cells by downregulating MMP-2 expression and ERK and JNK phosphorylation [29]. These findings are in line with the anti-metastatic effects of picrasidine J mediated via ERK signaling downregulation (Figure 6 and Figure 7).

The EMT process is characterized by the loss of epithelial features and gain of mesenchymal features, which make cancer cells more invasive and metastatic [22,30]. There is evidence indicating that an induction of EMT occurs following cancer treatments, which in turn increases the enrichment of chemotherapy-resistant cancer stem cells [31]. These observations highlight the significance of EMT in inducing cancer progression, recurrence, and treatment resistance. Several cancer therapeutics targeting EMT have been developed to improve cancer prognosis [32,33]. In the present study, we have clearly observed the EMT-suppressing effects of picrasidine J characterized by an induction in the expression of epithelial markers and a reduction in the expression of EMT activators (Figure 3). To the best of our knowledge, this study is the first of its kind to demonstrate the EMT-inhibiting effects of the picrasma-quassioides-derived compound picrasidine J.

Another novel observation made in this study is the picrasidine J-mediated downregulation of the serine protease KLK-10 (Figure 4). Specifically, we have observed that picrasidine J exerts anti-metastatic and anti-invasive effects by downregulating KLK-10 (Figure 5). As mentioned in the “Results” section, KLK-10 plays a crucial role in HNSCC onset and progression [26]. One previous study has shown that lower expression of microRNA-375 (miRNA-375) is associated with HNSCC recurrence, metastasis, and poor survival. The study has also shown that experimental induction of miRNA-375 expression results in reduced invadopodia formation and extracellular matrix degradation. As explained in the study, higher miRNA-375 expression leads to reduced secretion of commonly regulated proteases including KLK-10, which in turn is responsible for the suppression of extracellular matrix degradation and metastasis [24]. These observations support our findings about the association between picrasidine J-mediated KLK-10 downregulation and suppression of HNSCC metastasis. In this context, previous studies have shown that aberrant expression of human tissue kallikreins in many cancer types is associated with higher risk of distant metastasis and that these proteases induce cancer cell migration by inducing other proteases and degrading extracellular matrix proteins [34,35,36]. Our research findings indicate that picrasidine J effectively reduces the expression of KLK-10 and phosphorylated ERK. The existing literature has emphasized that the amplification of the EMT and FAK/SRC/ERK axes may contribute to the metastasis of pancreatic ductal adenocarcinoma by upregulating KLK-10 expression [37]. Given this context, we knew that the limitation of present study is lacking to explore the correlation between the ERK pathway and KLK-10 in HNSCCs. Further study should address this issue. Furthermore, our upcoming study will delve into the anti-metastatic effects of picrasidine J through animal experimentation.

## 4. Materials and Methods

### 4.1. Cell Culture

Two human HNSCC cell lines Ca9-22 and FaDu were selected for the experimentation. The Ca9-22 and FaDu cell lines were purchased from the Japanese Collection of Research Bioresource Cell Bank (Shinjuku, Japan) and cultured in Dulbecco’s Modified Eagle Medium (DMEM) (Life Technologies, Grand Island, NY, USA) supplemented with 10% fetal bovine serum (FBS), 10,000 U/mL of penicillin, and 10 mg/mL of streptomycin. All cell lines were maintained at 37 °C in a humidified atmosphere with 5% CO_2_.

### 4.2. Picrasidine J Treatments

Picrasidine J (purity: ≥98%) was obtained from ChemFaces (Wuhan, Hubei, China). The stock solution (100 mM) was prepared using dimethyl sulfoxide (DMSO, Sigma-Aldrich Co., St. Louis, MO, USA) and stored at –20 °C. The DMSO concentration was maintained at less than 0.2% for each experiment. For the treatment with picrasidine J, appropriate amounts of stock solution were administered to the medium to obtain the final experimental doses.

### 4.3. MTT Assay

To study the effects of picrasidine J on cell viability, an MTT (3-(4,5-dimethylthiazol-2-yl)-2,5-diphenyltetrazolium bromide, Sigma-Aldrich) assay was performed. Briefly, the cells were seeded onto 96-well plates and treated with different concentrations (0, 25, 50, or 100 μM) of picrasidine J for 24 h at 37 °C. Medium-diluted MTT was then added to each well, and the cells were incubated for 2 h at 37 °C. The formazan crystals were dissolved in DMSO and measured with a microplate reader (BioTek; Winooski, VT, USA) at 570 nm.

### 4.4. Wound Healing Assay

The cells were seeded onto 6-well plates and cultured to 90% confluence. The cell monolayer in each well was scratched with a 200 µL micropipette tip and treated with different doses (0, 25, 50, and 100 μM) of picrasidine J for 24 h. To quantify the migrated distance, the cells were photographed at 3, 6, and 24 h under a microscope. The migrated distance was determined with cell-free area using ImageJ, and the mean value was used for the analysis.

### 4.5. Cell Migration and Invasion Assay

Migration and invasion assays were performed as previously described [38]. After the treatment with different doses (0, 25, 50, and 100 μM) of picrasidine J for 24 h, the cells were seeded onto the upper chamber of culture inserts (Greiner Bio-One, Frickenhausen, Germany) with (for invasion assay) or without (for migration assay) Matrigel (BD Biosciences, Billerica, MA, USA). The chambers were placed in 24-well plates containing complete medium. After 24 h, the cells were fixed with methanol for 10 min and then stained with 10% Giemsa staining solution (Sigma-Aldrich). The stained cells were observed under a microscope for the measurement of their migration and invasion ability. The migrated and invaded cells were quantified using Image J (1.54d).

### 4.6. Western Blot Assay

Protein samples were extracted from the picrasidine J-treated cells with lysis buffer and separated using 10% polyacrylamide gel before transfer to polyvinylidene fluoride (PVDF) membranes (Merck Millipore). The membranes were then blocked using 5% nonfat milk prepared in a TBST buffer for 1 h and subsequently incubated for 24 h at 4 °C with primary antibodies against total and phosphorylated extracellular signal-related kinase (ERK), AKT, p38, c-Jun N-terminal kinase (JNK), β-actin, and KLK-10 (Thermo Fisher Scientific; Waltham, MA, USA). Subsequently, the cells were incubated for 1 h with secondary (peroxidase-conjugated) antibodies at room temperature. Finally, the protein bands were assessed using an ImageQuant LAS 4000 Mini (GE Healthcare Life Sciences; Boston, MA, USA).

### 4.7. Gene Silencing and Transfection

The KLK-10 siRNA and siRNA negative control were obtained from Cohesion Biosciences (London, UK). The cells were seeded onto a 6 cm dish and transfected with 25 nM siRNAs using TurboFect Transfection Reagent (Thermo Fisher Scientific; Waltham, MA, USA) for 24 h, then co-treated with or without picrasidine J for another 24 h. The status of KLK-10 downregulation was measured using Western blot assay.

### 4.8. Statistical Analysis

All statistical analyses were performed with SigmaPlot v12.5 (Systat Software Inc.; Palo Alto, CA, USA). The data were collected from three independent experiments. One-way analysis of variance and Tukey’s multiple comparison test were performed to compare the treated and non-treated control cells. A *p* value of <0.05 was considered statistically significant.

## 5. Conclusions

The present study demonstrates the anticancer effects of the *Picrasma quassioides-*derived compound picrasidine J. The compound exhibits high potency in inhibiting the motility, migration, and invasion of HNSCC cells by targeting the EMT pathway, KLK-10 expression, and ERK signaling.

## Figures and Tables

**Figure 1 ijms-24-13230-f001:**
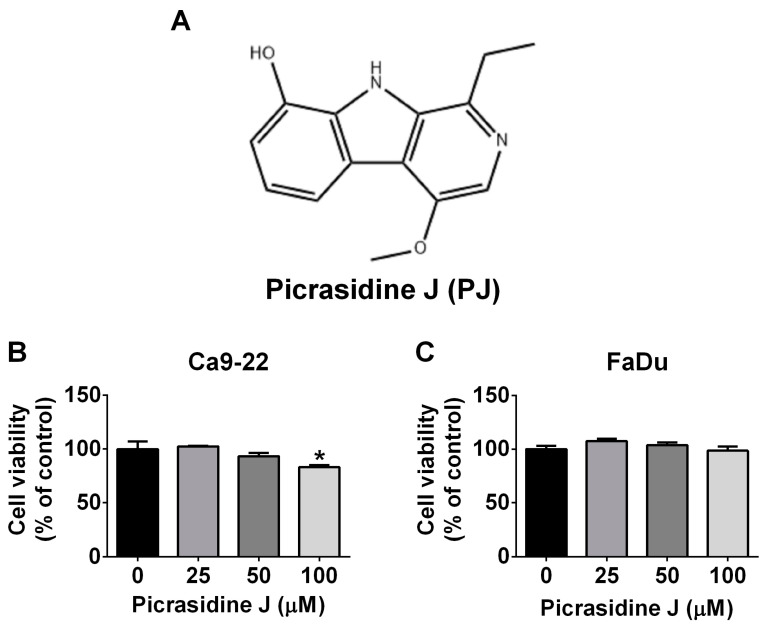
Toxicity of picrasidine J on HNSCC cancer cells. (**A**) Chemical structure of picrasidine J. (**B**,**C**) Results of the MTT assay for Ca9-22 and FaDu cells treated with indicated concentrations of picrasidine J (0, 25, 50, and 100 μM) for 24 h. Data are mean ± standard deviation (SD) of three independent experiments. * *p* < 0.05, compared to vehicle.

**Figure 2 ijms-24-13230-f002:**
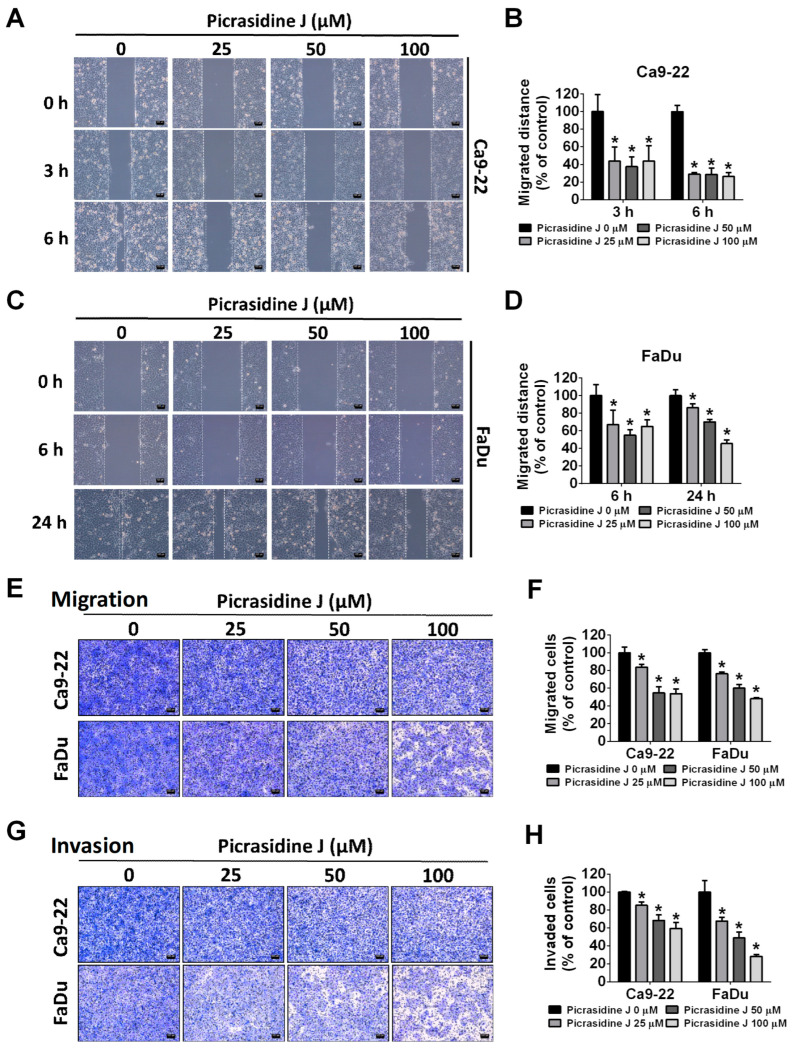
Picrasidine J inhibits horizontal migration and invasion of HNSCC cancer cells. (**A**,**B**) Ca9-22 and (**C**,**D**) FaDu cells treated with indicated concentrations of picrasidine J (0, 25, 50, and 100 μM) and photographed at 0, 3, 6, and 24 h to measure the distance migrated. (**E**,**F**) Migration and (**G**,**H**) invasion assays for Ca9-22 and FaDu cell lines. Cells stained with Giemsa staining buffer were counted to analyze cell migration and invasion. Data are mean ± SD of three independent experiments. * *p* < 0.05, compared to vehicle. Scale bar: 100 μm.

**Figure 3 ijms-24-13230-f003:**
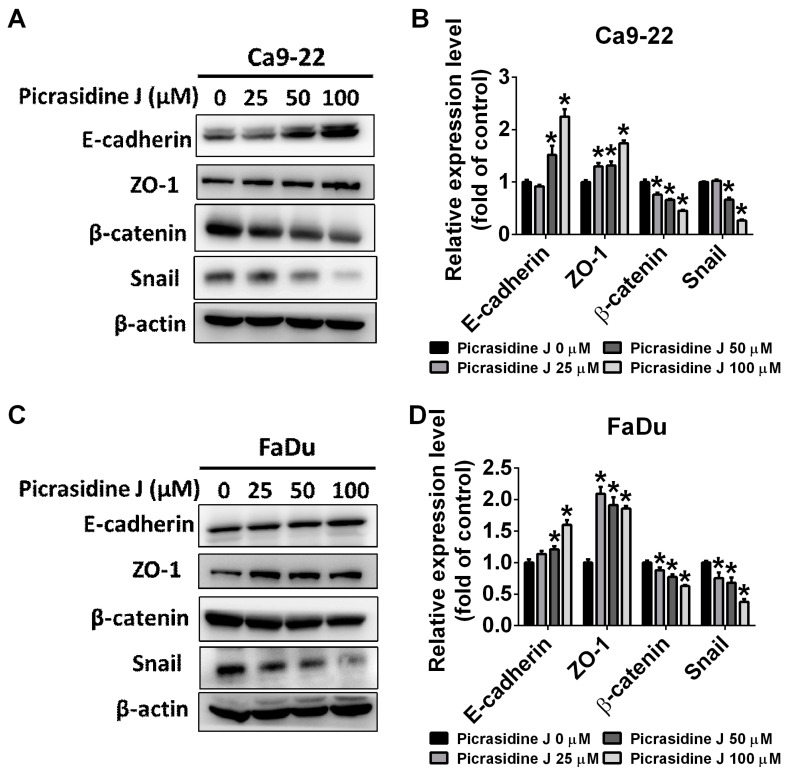
Picrasidine J inhibits the expression of epithelial–mesenchymal transition (EMT)-related proteins in HNSCC cancer cell lines. After picrasidine J treatment, E-cadherin, ZO-1, β-catenin, and Snail levels in (**A**,**B**) Ca9-22 and (**C**,**D**) FaDu cells were measured using Western blot assay with β-actin as an internal control. Data are mean ± SD of three independent experiments. * *p* < 0.05, compared to vehicle.

**Figure 4 ijms-24-13230-f004:**
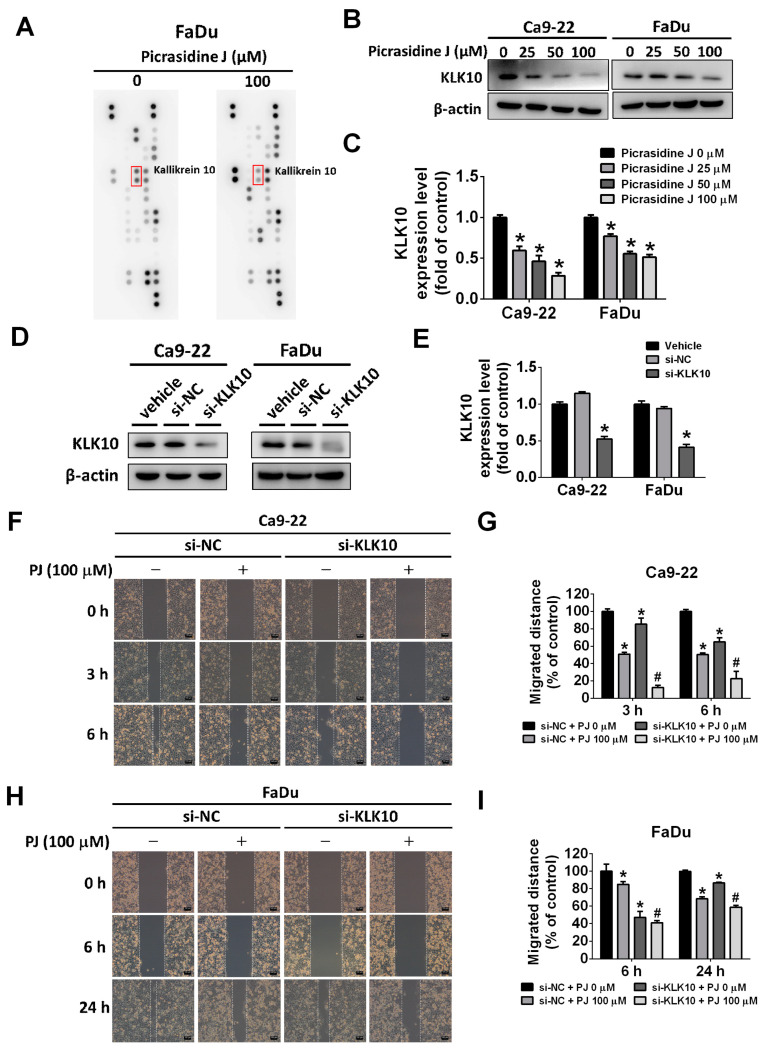
Picrasidine J regulates cell metastasis by downregulating protease KLK-10 in HNSCC cancer cells. (**A**) Cells treated with 100 μM of picrasidine J were collected and measured metastatic-related protein expression using Human Protease Array Kit (Cat. ARY021B, R&D Systems Inc.). (**B**,**C**) KLK-10 measured using Western blot assay after picrasidine J treatment (0, 25, 50, or 100 μM), with β-actin as internal control. (**D**,**E**) KLK-10 level measured after transfection with si-NC (negative control siRNA) or si-KLK-10 siRNA using a Western blot assay, with β-actin as internal control. (**F**,**G**) Ca9-22 and (**H**,**I**) FaDu cells were transfected with si-NC or si-KLK-10 siRNA after treatment with or without picrasidine J (100 μM). After 0, 3, 6, and 24 h, a wound healing assay was used to analyze Ca9-22 and FaDu cells. * *p* < 0.05; ^#^
*p* < 0.05, compared to picrasidine J only. Scale bar: 100 μm.

**Figure 5 ijms-24-13230-f005:**
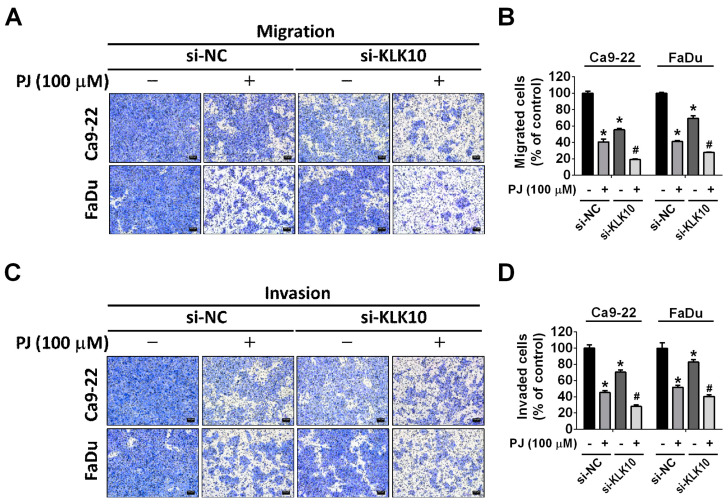
KLK-10 participates in picrasidine J-induced metastasis. (**A**,**B**) Cells were transfected with si-NC or si-KLK-10 siRNA following treatment with or without picrasidine J (100 μM) for 24 h and subjected to migration and (**C**,**D**) invasion assays. Data are mean ± SD of three independent experiments. * *p* < 0.05, compared to vehicle; ^#^
*p* < 0.05, compared with picrasidine J only. Scale bar: 100 μm.

**Figure 6 ijms-24-13230-f006:**
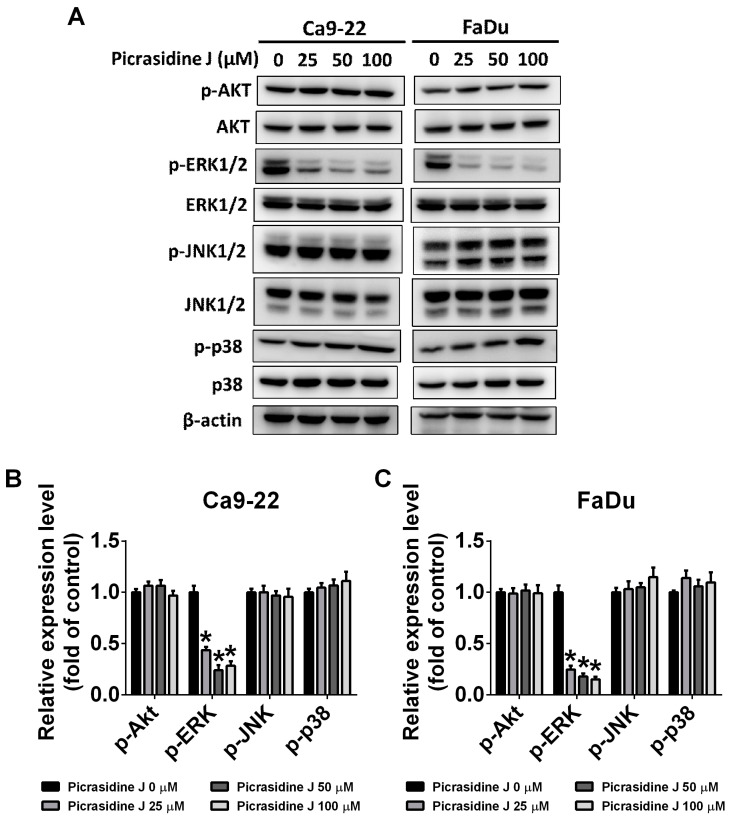
MAPK pathways involved in picrasidine J-induced metastasis regulation in HNSCC cancer cells. (**A**–**C**) After picrasidine J treatment, total and phosphorylated AKT, ERK, p38, and JNK levels in Ca9-22 and FaDu cells were measured using a Western blot assay with β-actin as an internal control. Each phosphorylated form of a protein is quantified in relation to its total form’s level. Data are mean ± SD of three independent experiments. * *p* < 0.05, compared to vehicle.

**Figure 7 ijms-24-13230-f007:**
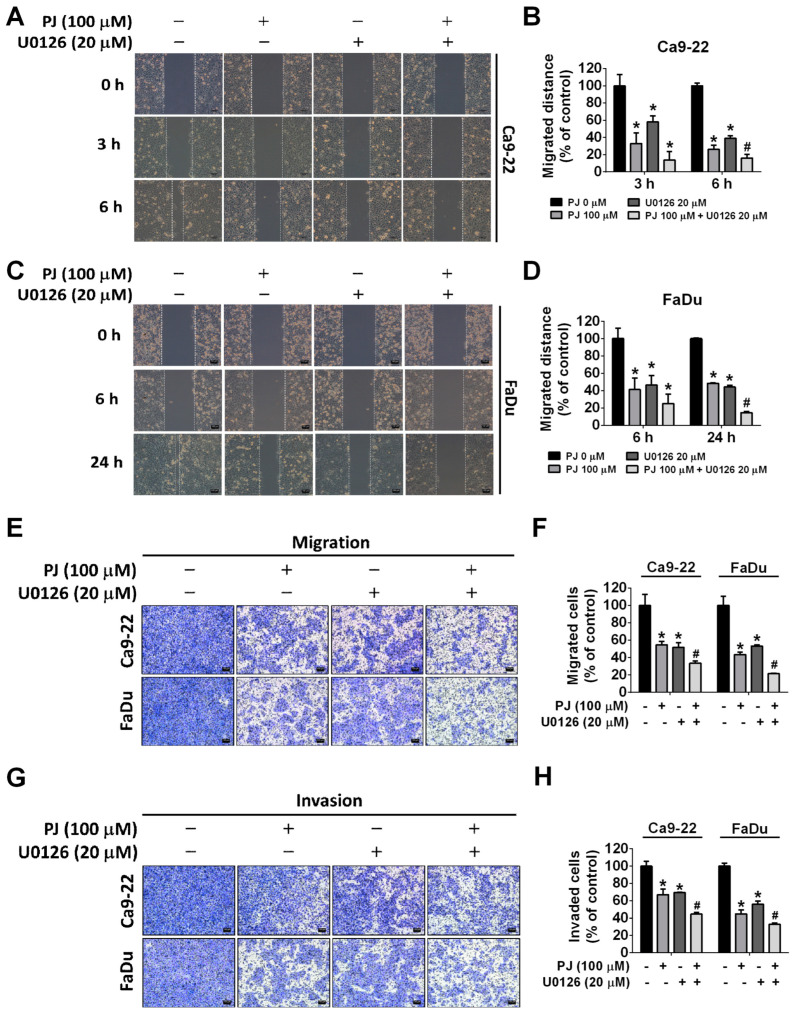
ERK pathway participates in picrasidine J-induced metastasis regulation. (**A**,**B**) Ca9-22 and (**C**,**D**) FaDu cells treated with the ERK inhibitor (U0126) for 1 h. Cells were cotreated (or not) with 100 μM of picrasidine J and analyzed using a wound healing assay after 0, 3, 6, and 24 h. Data are mean ± SD of three independent experiments. * *p* < 0.05; ^#^
*p* < 0.05, compared to picrasidine J only. Ca9-22 and FaDu cells were treated with the ERK inhibitor (U0126) for 1 h. Cells were cotreated (or not) with 100 μM of picrasidine J for 24 h and subjected to migration (**E**,**F**) and (**G**,**H**) invasion assays. Data are mean ± SD of three independent experiments. * *p* < 0.05, compared to vehicle; ^#^
*p* < 0.05, compared with picrasidine J only. Scale bar: 100 μm.

## Data Availability

The data used to support the findings of this study are available from the corresponding author upon request.

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
