# Peer review of "Picrasidine J, a Dimeric β-Carboline-Type Alkaloid from Picrasma quassioides, Inhibits Metastasis of Head and Neck Squamous Cell Carcinoma"

_ijms, 2023, doi:10.3390/ijms241713230_

Round 1

Reviewer 1 Report

This manuscript reveals anti-metastatic efects of compound picrasidine J against HNSCC cells. The strong study design includes co-treatment of PJ with siRNA knockdown specific targeted molecule KLK10 and ERK inhibitor to clarify related signaling molecules interacting PJ action. 

Most importantly, the significance and importance of picrasidine, as well as originated plants, for health benefit explained in the manuscript is not enoung to convinct readers to gain novel finding. Author must deeply explain in background of this compound and plants to convinct as beneficial and potential use for health.

I have some questions and suggestions regarding background and result interpretation:

1. Introduction part presented effects of other types of picrasidine on cancers. But no previous findings on picrasidine J. What's the rationale of PJ that interesting in this study? And what are diiferences from other types of picrasidine that has been reported previously?

2. For cytotoxic testing, DMSO as solvent for PJ should be tested and compared with untreated group. Section 2.1 stated that the cells without any treatment were used as controls, please clarify.

3. Caption of figure 4A must be added.

4. By results of wound healing in figure 4F, the width area of siNC+PJ and si-KLK10-PJ in Ca9-22 at 3h and 6h are seem equal each other at the same timepoint. But the bar graph showed significant difference between them at the same timepoint. Please re-analyze or explain.

5. By immunoblotting results of MAPK proteins, please clarify that the relative expression level of each protein were quantified in relative level compared with each total form or b-actin?

6. Please refer or explain the rationale of timepoint selection for treatment of U0126 for 1 h.

7. How to measure or count migrated and invaded cells following migration/invasion assay. Also, the stained cells after the assay is live or fixed? Please add in material and method section.

8. Please explain concentration and time-point condition of transfection.

9. Please explain limitation in this study and expected further studies.

10. References is redundant (42 list of references). Please reduce or cite only potential or significant references. Some typical method should not be cited.

Most of spelling is good, not detect any misspelling. But some punctuation error and mis-spacing are obvious. Please carefully re-check.

Reviewer 2 Report

The authors present data supporting anti-cancer effect of a natural compound using in vitro assays. they have used two human HNSCC cell lines and multiple combinations for enhancing the anti-cancer effect of picrasidine J.

The authors need to revise their manuscript to explain the results better as the effect observed with combination of picrasidine j with siKLK10 or ERK inhibitor is more like an additive effective suggesting that picrasidine has multiple effects. they have not included the siKLK10+ERK inhibitor and compared the effects to suggest that only these two molecules are the targets for picrasidine j in these HNSCC cells.

minor points:

the authors need to change the title of the results subheadings to reflect the results instead of methods

for example

2.5. Effect of Picrasidine Jon MAPK signaling pathwayin HNSCC cells.

this can be rephrased "picrasidine J inhbits ERK activation in HNSCC cells"

also the heading for figure 5 is incorrect

Figure 5. KLK10 participating in Picrasidine J-induced metastasis.

there are some spelling and grammar mistakes.

see above

Round 2

Reviewer 1 Report

All coorections and suggestions are responsed by author and suitable for publication.